# Porous translucent electrodes enhance current generation from photosynthetic biofilms

Tobias Wenzel [1], Daniel Härtter [1,2], Paolo Bombelli[3], Christopher J. Howe[3] & Ullrich Steiner [4]

Some photosynthetically active bacteria transfer electrons across their membranes, generating electrical photocurrents in biofilms. Devices harvesting solar energy by this mechanism are currently limited by the charge transfer to the electrode. Here, we report the enhancement of bioelectrochemical photocurrent harvesting using electrodes with porosities on the nanometre and micrometre length scale. For the cyanobacteria *Nostoc punctiforme* and *Synechocystis* sp. PCC6803 on structured indium-tin-oxide electrodes, an increase in current generation by two orders of magnitude is observed compared to a non-porous electrode. In addition, the photo response is substantially faster compared to non-porous anodes. Electrodes with large enough mesopores for the cells to inhabit show only a small advantage over purely nanoporous electrode morphologies, suggesting the prevalence of a redox shuttle mechanism in the electron transfer from the bacteria to the electrode over a direct conduction mechanism. Our results highlight the importance of electrode nanoporosity in the design of electrochemical bio-interfaces.

[1] Cavendish Laboratory, Department of Physics, University of Cambridge, Cambridge CB3 0HE, UK. [2] III. Physikalisches Institut, Georg-August-Universität, 37077 Göttingen, Germany. [3] Department of Biochemistry, University of Cambridge, Hopkins Building, Cambridge CB2 1QW, UK. [4] Adolphe Merkle Institute, Rue des Verdiers 4, 1700 Fribourg, Switzerland. Correspondence and requests for materials should be addressed to C.J.H. (email: ch26@cam.ac.uk) or to U.S. (email: ullrich.steiner@unifr.ch)

Several microorganisms are able to generate electrons that can be collected and utilised in external circuits[1]. In such devices, bio-anodes are the electrodes that collect electrons from the living bio-catalyst. Bio-anodes in the best studied bioelectrochemical technology, microbial fuel cells, are commonly carbon or metal based, and a large diversity of morphologies has been used[2]. The electrode porosity usually has a strong effect on device efficiency[2], but the associated change in volume, surface area and organism contact area can rarely be disentangled from the variation of materials themselves that are used to achieve the different morphologies. These complicated correlations currently limit the understanding of design rules for electrochemical bio-interfaces. Furthermore, because of a lack of transparency of most anodes, there has been little work on the benefits of using porous electrodes in microbe-based devices that rely on light absorption, referred to as 'biophotovoltaics', except for one study using larger, eukaryotic, algal cells[3]. Although photosynthetic microorganisms are expected to operate with a quantum efficiency of five to ten percent internally, electrode interfaces and microbial electron export pathways currently limit device efficiencies to much lower values[1].

In this study, we tested the effect of electrode porosity at different length scales on the performance of bioelectrochemical devices. To achieve this goal, we have compared three different electrode morphologies of the same translucent material. Two photosynthetic microorganisms Nostoc punctiforme and Synechocystis sp. PCC 6803 were each placed on a non-porous indium tin oxide (ITO) electrode, a thick 'nanoporous' ITO nanoparticle film, and a 'microporous' inverse-opal structure made from the same nanoparticles, and their photocatalytic current generation was investigated. Doped metal oxides are popular transparent electrode materials for a wide range of electronic applications. ITO is one of the best performing transparent electrode materials, as it has a large optical bandgap, making it transparent to visible light, while the high levels of tin doping cause a metal-like conductivity. ITO is commonly used as thin film (tens of nanometres thick) in display applications and was shown to be biocompatible[4–6]. ITO can be structured using a templating approach and has previously been used as porous glass in electrochemical studies of enzymes[7, 8]. In order to distinguish between the porosity on different length scales, we define nanoporosity as the presence of pores between sintered nanoparticles (10–100 nm), and microporosity as the pores created by microsphere templates (10–40 μm). The templated inverse-opal pores used in this study are unusually large and the electrodes are unusually thick (80–140 μm) in order to accommodate a sufficient number of microorganisms within the structure to absorb incoming light. The two porosity length scales were chosen to represent biologically relevant regimes of electron transfer from microorganisms to the anode. The extracellular electron export mechanisms in cyanobacteria are still unclear, even for the model organism Synechocystis[1, 9]. Research on microbial electrochemical devices distinguishes between direct electron transfer (DET) from microorganisms to electrodes, and mediated electron transfer (MET) facilitated by electrochemically active molecules in solution. Nanopores are not directly accessible by the relatively large microbes, but the increased surface area is available for electrochemical interactions with redox-molecules in the aqueous electrolyte, which is relevant for MET pathways. Pore sizes comparable to the cell size in microporous morphologies allow cells to enter the electrode, thereby providing a considerable increase in direct contact area between bacteria and electrode surface.

Biophotovoltaic devices generate electric energy through photo-electrons generated by living photosynthetic organisms on the anode, and a cathodic reaction taking place at a lower

potential, here recombination to water. The energy levels involved in this process are detailed in the schematic diagram in Fig. 1. The starting point of photosynthesis is photocatalytic water splitting, during which the electrons are gained. This is achieved through the absorption of two photons by the two photosystems (PSII and PSI), which form the characteristic Z-Scheme. The derived electrons may be stored through the production of NADPH molecules and can leave the cells via extracellular electron transfer (MET & DET). The literature suggests similar values for redox-potentials of electron export in known organisms[10], but with different efficiencies[11]. Taking interfacial energy transfer losses into account, these potentials determine the maximum voltage that can be generated in the device, vs. the fixed potential of a platinum air-cathode. In practice, the cathode potential of air cathodes does not reach the theoretical level of water splitting[10], which is indicated by the higher position of the cathode level in the scheme.

The two cyanobacteria studied here, Nostoc punctiforme and Synechocystis sp. PCC 6803 exhibit a remarkable ca. 300 fold increase in generated peak photocurrent when cultured on nanoporous and microporous bio-anodes, compared to non-porous ITO films that are routinely used for biophotovoltaic devices. The non-photosynthetic microorganism Shewanella oneidensis shows a similarly dramatic increase in external current generation on the two porous electrode types.

## Results

**Electrode design and characterisation.** A porous electrode that can incorporate photo-active biofilms must combine the three qualities, conductivity, translucency and microporosity on a length scale that allows cells to enter interconnected pores, while also forming biofilms with a thickness up to millimetres. To this end, an inverse-opal structure was designed with pores and pore-connections of several micrometres, similar to that of cyanobacterial cells. The inverse-opal structure was generated through a templating approach (Fig. 2a). Polymer microspheres of 40 μm in diameter were deposited to form an opal structure, which was annealed to promote sphere adhesion and to control the diameter of the sphere-sphere interconnects, followed by infiltration by an ITO nanoparticle suspension via capillary forces. Filling the

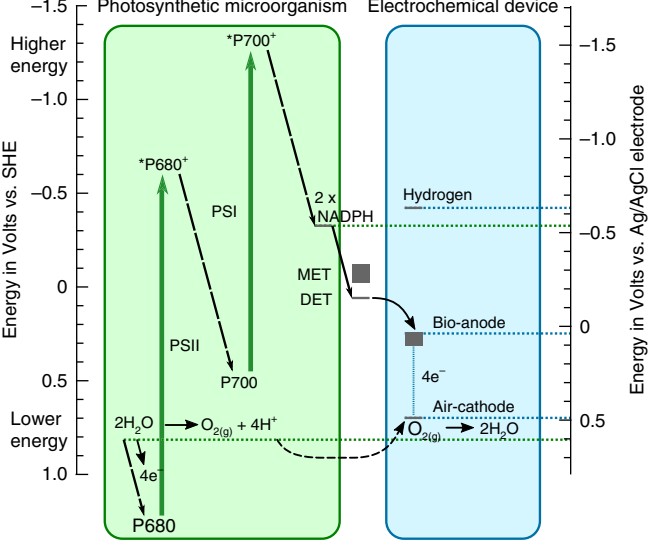

**Fig. 1** BPV energetic scheme. Energy levels (at physiological pH) inside photosynthetic organisms and charge transfer to external electrodes of biophotovoltaic devices

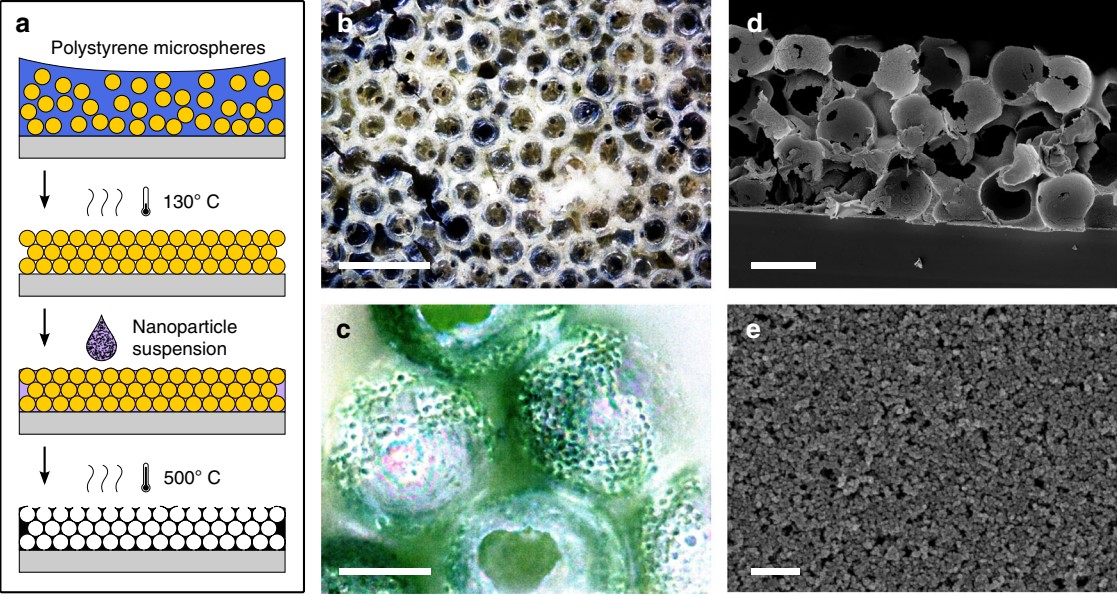

**Fig. 2** Electrode porosity. **a** Scheme of microporous electrode fabrication method. **b** Optical bright field microscopy image of an empty microporous electrode (scale bar: 100 μm). **c** Optical dark field microscopy image showing an electrode filled with cyanobacteria (after rinsing with water; scale bar: 20 μm). Scanning electron microscopy (SEM) images showing **d** the cross-section of the electrode (scale bar: 40 μm) and **e** the nanoporous structure of sintered nanoparticles (scale bar: 400 nm)

template with nanoparticles (<100 nm) proved to be the fastest and most reliable method to obtain thick porous films. A final heating step sintered the nanoparticles, while burning out the polymer template to leave the inverse structure behind.

The resulting material was diffuse white-yellow, as seen by eye or optical microscopy (Fig. 2b). After the addition of cyanobacterial cells, the electrode was examined by light microscopy (Fig. 2c), to confirm the ability of the cells to populate the structure across the entire thickness (here, ca. 0.12 mm). The cells (ca. 2 μm cell-diameter) had to pass through ca. 10 μm wide connections in order to reach all hollow spheres of the inverse-opal morphology (see Fig. 2b and cross-section in d) or reach these spaces via cracks. The nano sized pores of the electrode material (Fig. 2e) could not be accessed by the bacteria.

To compare the effects of microporosity, nanoporosity and no porosity in this study, three different ITO structures were used. A commercial non-porous ITO layer on a PET substrate served as reference (Supplementary Fig. 1a, b), and thick nanoparticle films (Fig. 2e, Supplementary Fig. 1c–f) without and with additional micropores (Fig. 2d, Supplementary Fig. 1g, h) were employed to assess the interplay of nanoporosity and microporosity on the charge generation by the microorganisms.

The nanoparticle film had a thickness of ca. 9 μm and was produced with the same nanoparticles and on the same conducting FTO-glass substrate as the microporous electrodes. Its sheet resistivity on a non-conducting substrate was ca. 100 Ω cm$^{-2}$, determined by a 4-point probe. The film resistivity was determined by impedance spectroscopy (see Methods). The resistivity of dry electrode-'sandwich'-samples with an area of approximately 0.25 cm$^2$ were measured as a function of film thickness. The obtained values were then compared to that of thicker inverse-opal structured electrodes, the conductivity of which may be limited by their 3D morphology. The electrodes exhibited purely ohmic resistance of 65 and 95 Ω for temperature annealed ITO glass and 10 μm thick sandwiched nanoparticle

films, respectively (Supplementary Fig. 2). The ca. 140 μm thick microporous electrodes displayed similarly low purely ohmic resistance of ca. 115 Ω.

The visual appearance and optical transmission spectra of the microporous and the nanoparticle film are shown in Supplementary Fig. 3. Ca. 50% of the visible spectrum was diffusely transmitted through the dry microporous film, and ca. 80% through the dry nanoparticle film. Non-porous ITO is fully transparent to visible light, but causes partial specular reflection at air or water interfaces due to its high refractive index. Nanoporous ITO (Fig. 2e) contains defects that scatter light, rendering electrodes translucent, which is even stronger for the internal interfaces of the inverse-opal structure (Fig. 2d). The relatively large opal unit cell size of 40 μm was chosen to minimise the number of metal oxide water interfaces, and, thus, the back-scattering of light, and to allow the use of large microorganisms. Scattering and absorption were more strongly pronounced in this electrode than in previously described ITO-based inverse-opal structures (e.g. in[12] ca. 80% of the visible spectrum is transmitted), due to the unusually large thickness of the electrode.

**Biophotovoltaic devices**. Figure 3 pictures the electrochemical devices used for photo electricity measurements in up to 12 parallel channels. The devices were 3D-printed to allow for a compact, shareable design and to enable temperature control via hollow walls and a circulating water bath. The structured anodes on FTO-glass were inserted from the bottom on a printed anode holder, and sealed with an O-ring. The devices were designed to leave a (macroscopic) electrode area of 1 cm$^2$ exposed to micro-organisms and light. Three pieces of platinum nanoparticle based air cathodes with a combined area of ca. 6 cm$^2$ were arranged upright and close to each anode opening. Parallel measurements with the same batch of bacteria culture and under the same high-

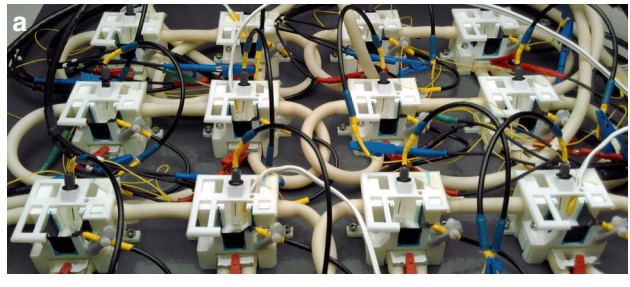

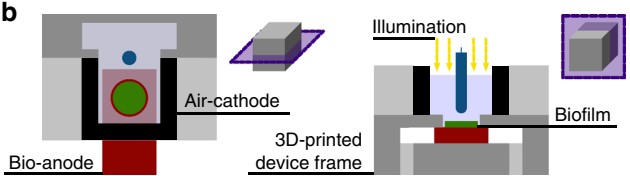

**Fig. 3** Design of biophotovoltaic devices. **a** Assembled and connected 3D-printed devices, incl. reference electrodes with with blue connectors, positioned in the homogeneous illumination area. **b** Two schematic perspectives of cuts through a device indicating the locations of electrodes and biofilm

intensity light source were essential to obtain reliable mean results in experiments.

**Photocurrent ratios**. To study the effect of electrode porosity on the biophotovoltaic device performance, parallel measurements with equal numbers of cells (based on amount of chlorophyll) were conducted on the three different electrode structures. The peak photocurrent levels observed with *Synechocystis* are shown in Fig. 4a. During the exposure of devices to light, the photocurrent did not stay constant, but reached a peak value, followed by a decreasing slope, slowly reaching a steady value. The peak photocurrent was defined as the average short-circuit current (0 V external potential between anode and cathode and no reference electrode) measured during one minute around its peak value, minus the dark-current level before each illumination, see Supplementary Fig. 5.

Both microporous and nanoporous bio-anode based devices substantially outperformed the non-porous ITO electrode with 11.5 and 8.4 mA m$^{-2}$ to 0.04 mA m$^{-2}$, respectively, while the microporous structure displayed a small, but consistent advantage over the film with nanopores only. During the illumination period of seven minutes, the microporous devices reached a peak photocurrent value ~ 300 times higher than those equipped with non-porous electrodes.

In half-cell operation, the total charges collected during a longer period of 90 minutes were larger by more than a factor of 10 with microporous vs. non-porous anodes, see Fig. 4b. Equilibrium values of the photocurrent are difficult to obtain because BPV generated currents tend to fluctuate over time. Photocurrent measurements are also limited by photo bleaching of photosynthetic pigments during illumination and the long time periods necessary for bio-anodes to stabilise after a change of conditions. To quantify the photovoltaic output of the devices, the number of electrons generated during 90 minutes of illumination (Fig. 4b) was therefore considered as a robust current measure in addition to the recorded peak values (Fig. 4a).

Similar results were obtained for *Nostoc* bio-anodes, where the current enhancement of devices with a microporous electrode

was also higher by a factor of ~300 compared to non-porous electrodes, with a peak current of 11.2 mA m$^{-2}$, and 30 times more charge collected during 90 minutes (167 μmol m$^{-2}$ s$^{-1}$).

Figure 4d illustrates the BPV device chamber at 'short-circuit' in which the cathodic water recombination reaction drives the device current and voltage, without an additional force of an externally applied potential. Electrochemical studies commonly use a half-cell configuration instead (Fig. 4e). There, the anode potential is set with respect to a reference electrode of a stable and known potential, here 0.2 V vs. an Ag/AgCl reference electrode, and the cathode is dynamically shifted to a potential, where the cathodic reaction is non-limiting for the measurement. To test whether the cathode limits the peak photocurrents reached in our devices, a set of peak and continuum measurements were performed in half-cell mode, shown alongside the device measurements in Fig. 4a, b. The highest current levels observed, here in the case of porous anodes, were consistent between device and half-cell operation within each others standard deviations, indicating the absence of cathode limitation.

For comparison with an organism of well-studied ability to perform DET[13], an additional test was performed with non-photosynthetic *Shewanella oneidensis* bacteria in the three electrode set-up. The average generated currents were 0.5 mA m$^{-2}$, 62 mA m$^{-2}$ and 299 mA m$^{-2}$ for non-porous, nanoporous and microporous bio-anodes, respectively (Fig. 4c), recorded in half-cell mode at an anode potential of 0.2 V vs. a Ag/AgCl reference electrode in fresh LB medium. This corresponded to a more than 100-fold current increase from non-porous to nanoporous electrodes, and a further ca. 5 times increase for the microporous structure.

**Light response characteristics**. Exposing photosynthetic electroactive biofilms to light gave rise to an initial current peak, which typically dropped to steady-state values. For each of the organisms studied here, the photocurrent rose much faster to its peak value on devices employing one of the porous compared to non-porous electrodes, with little difference between the two pore types (Fig. 5 and Supplementary Fig. 6). The current peaks were reached for porous electrodes after 1–6 minutes, whereas devices with non-porous electrodes required up to over one hour of light exposure to reach a maximum (Fig. 5 and Supplementary Fig. 6 expanded regions). The steady-state current reached by non-porous bio-anodes was still lower compared to the porous electrodes. Note that the photocurrent minimum for *Synechocystis* cells on a non-porous electrode (Fig. 5) is unexpected. It was not observed when the measurement was carried out with reference electrode (Supplementary Fig. 6). Since this study is concerned with the current maximum which was reached after two hours, the origin of the minimum was not further investigated.

The fast photo response of the porous bio-anodes enabled serial experiments studying the peak BPV photocurrent for different light irradiances, see Fig. 6a, b. The resulting curves show an approximately exponential saturation of photoelectron generation, with higher efficiencies (ratio of photocurrent to photon flux) reached at low light levels. The performance advantage of porous electrodes persisted for all irradiation levels. The similarity of photocurrents on porous electrode types when used with either *Synechocystis* or *Nostoc* indicates the importance of the effective mesoscopic electrode surface area rather than its coarse morphology. At the same time, the similar saturation curves for each of these morphologies confirms the expectation that the microporous electrode architecture does not shade cells to a significant degree.

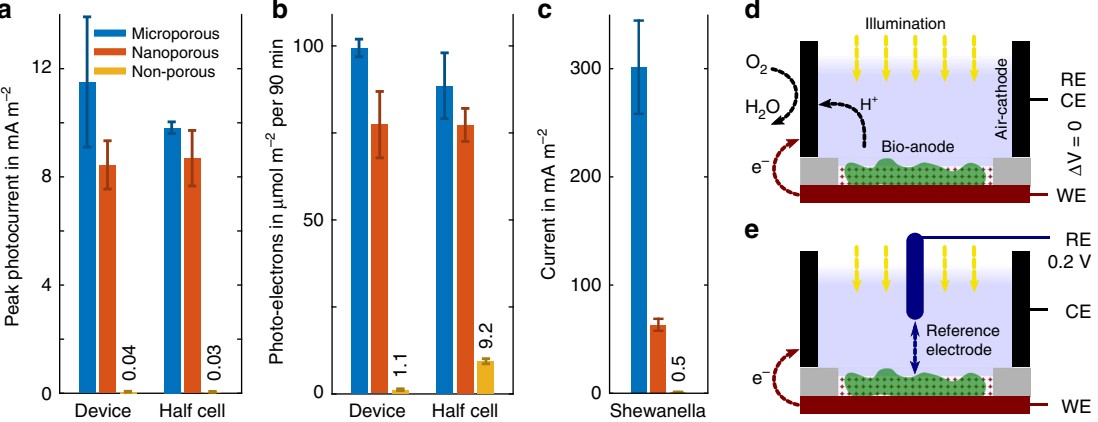

**Fig. 4** Current levels by anode type and device configuration. **a** *Synechocystis* peak photocurrents in device mode (at 'short-circuit') and half-cell operation (with reference electrode), during a photon flux of 460 µmol m$^{-2}$ s$^{-1}$. **b** As (**a**), number of collected charges during 90 minutes of illumination at photon flux of 512 µmol m$^{-2}$ s$^{-1}$. **c** Current generated by Shewanella cells for each anode type. **d**, **e** Schematic illustrating the device and half-cell operation, respectively. The potentiostat connections for reference (RE), working (WE) and counter electrode (CE) are indicated

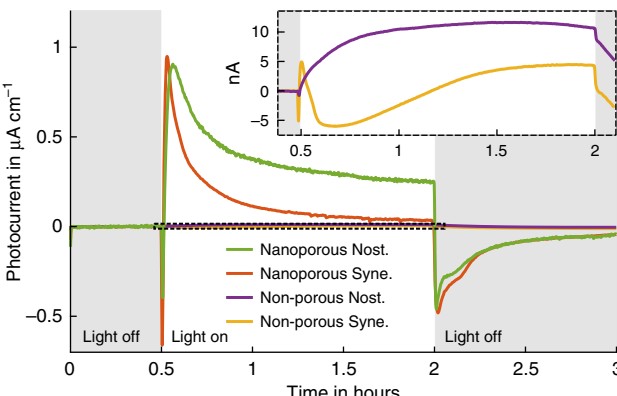

**Fig. 5** Light response dynamics. Development of the average BPV photocurrent over time at a photon flux density of 512 µmol m$^{-2}$ s$^{-1}$, for non-porous and nanoporous film electrodes and both *Nostoc* and *Synechocystis*

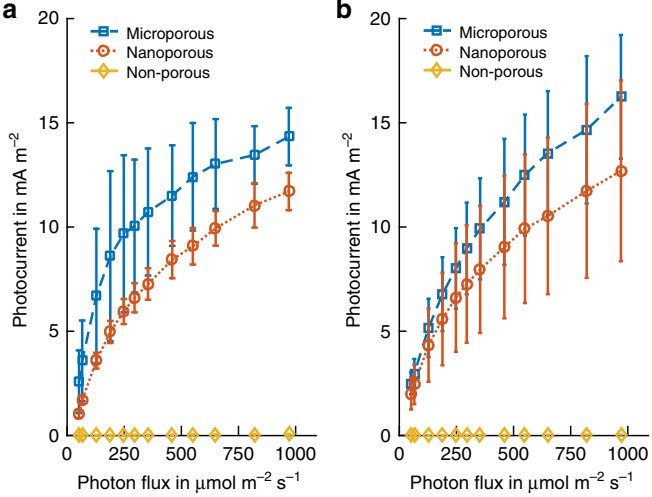

**Fig. 6** Saturation behaviour. **a** Peak current generated by *Synechocystis* cells on the three anode types at different light levels. The values are averages of the peak minute after subtracting the dark current. **b** *Nostoc* biofilm maximum photocurrents as in (**a**)

In addition to the use of porous bio-anodes, the need for fast and accurate measurements was addressed by adding phosphate buffer to the BG11 medium, which increased the electrolyte conductivity, by disregarding preceding dark-current levels, and by choosing dark-times between (short) illuminations that were long enough for the dark-current level to recover. For the low light level *Nostoc* measurements (photon flux 0–400 µmol m$^{-2}$ s$^{-1}$, Fig. 6b), the irradiation intervals were not long enough to reach the full peak values, which may explain the lower exponential slope for low photon fluxes compared to the *Synechocystis* data.

Surprisingly, in contrast to porous electrodes, non-porous ITO electrodes performed better and responded faster in the absence of phosphate buffer compared to relative performance in the presence of buffer, see Supplementary Fig. 7. However, their performance remained much below that of the microporous electrodes, which collected ten times more electrons during a 20 min illumination period.

**Redox reactions at the bio-anode.** The activity of electron-generation by bio-catalytic water splitting depends only on the input energy of the illuminating light source (Fig. 1). By

measuring the electron collection as a function of anode potential by cyclic voltammetry (CV), the redox activity of downstream electron donating molecules can be characterised[14]. The bio-catalytic activity is slow for most organisms, however, and could not be detected clearly by CV even at scan rates as low as 0.5 mV s$^{-1}$. The porous anodes showed a strong enhancement in electrochemical sensitivity, but also stronger surface charging, leading to high charging currents (non-Faradaic) that broaden the CV hysteresis curve, and, thus, mask bio-catalytic peaks, see Supplementary Figs. 8 and 9. This charge accumulation is also reflected by the transient spikes observed in chronoamperometry measurements when the light is turned off (e.g. Fig. 5 and Supplementary Fig. 5).

The increased electrochemical sensitivity of microporous electrodes did, however, enable the detection of a light dependent reversible peak of a reduction reaction on the anode surface (Fig. 6a). This peak was not observed on the less sensitive non-porous ITO anodes. The slight shift in peak position with irradiance was probably caused by a pH shift of the BG11-

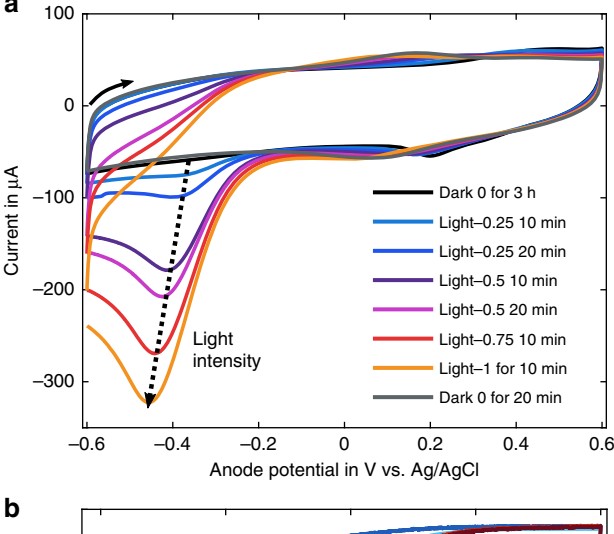

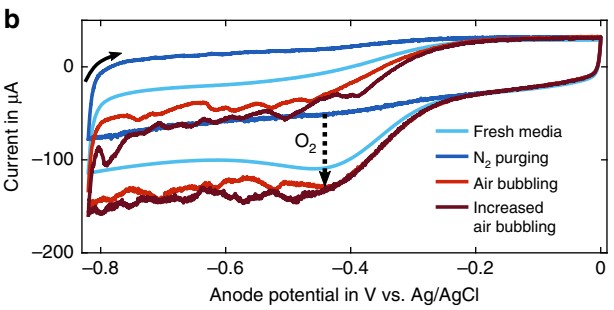

**Fig. 7** Cyclic voltammetry analysis of microporous ITO anodes at a scan rate of 4 mV s$^{-1}$ with BG11 medium as electrolyte. **a** Response of the bio-anode (*Synechocystis*) under exposure to different constant light levels. Light intensity 1 corresponds to a photon flux of 820 μmol m$^{-2}$ s$^{-1}$, and the time indicates for how long the light level was applied by the end of the measurement. **b** Scans of abiotic anodes in the dark. The oxygen content of the electrolyte was varied by purging the electrolyte with nitrogen gas or compressed air

medium electrolyte (no additional phosphate buffer was used in this measurement). Lowering the oxygen content of the electrolyte by purging it with nitrogen gas led to a decrease in peak height (Supplementary Fig. 10), suggesting its link to photosynthetic oxygen production. This was further confirmed by a measurement without microorganisms, during which the oxygen content was varied by purging with nitrogen or air, and for which the same reduction peak was reduced (nitrogen) and enhanced (air), see Fig. 7b. The reaction appeared at anode potentials lower than −300 mV, and was well separated from the operating range of presented bio-anodes, which had a measured electrode potential of 50 to 150 mV vs. Ag/AgCl (Fig. 1). This separation is important for device performance, as the electron-consuming reduction of oxygen can compete with the collection of photosynthesis-derived electrons (oxidation reaction) if it takes place at the material-dependent and pH-dependent operating potential of the bio-anode.

## Discussion

Translucent conductive electrodes with porosity on two relevant length scales—nanoporous films with pores accessible only by electrolyte and microporous electrodes with additional pores on the organism's length scale—were successfully created. Two cyanobacteria, *Synechocystis* sp. PCC 6803 and *Nostoc*

*punctiforme*, were applied to these electrodes where they continued to be electroactive for the duration of measurements (at least several days), with no indication of bleaching. During measurements, oxygen reduction at the anode is unlikely to compete with the collection of microorganism-derived electrons, as it takes place at lower anode potentials (ca. −400 mV) than those reached during BPV operation (ca. 100 mV).

In the architecture presented here, the cathode did not limit device performance (observed peak photocurrents), and half-cell measurements were equivalent to overall device performance. Measuring the bio-anode vs. a reference electrode (half-cell mode) simplifies the interpretation of results of the anode performance and the presence of a reference potential allows the application of methods, such as CV. However, half-cell measurements can only be related to external quantum efficiencies as defined in the field of solar energy generation, when it is known that an appropriate non-limiting cathode, such as the here-presented can be designed to complement it in a full device.

Photocurrents collected from both types of porous electrodes were typically one to two orders of magnitude larger than those from bio-anodes based on non-porous ITO electrodes. The performance advantage varied depending on the metric used (higher for peak currents; lower for charges collected over several hours) and electrolyte conductivity (higher salt levels improved the performance of porous anodes). Photocurrent peak values were also reached faster (up to 90 times) on porous electrodes. The similar photocurrent levels of both porous electrode types for each cyanobacteria species (Fig. 5) indicates that the current enhancement can be attributed to a non-limiting electrode surface area. The porous electrodes accordingly displayed similar coulombic currents caused by surface charging (Supplementary Figs. 8 and 9). Both high performance and fast response demonstrate the importance of a large electrode surface area for the design of electrochemical bio-interfaces.

The fast response of porous bio-anodes enabled the performance of timely serial experiments measuring the dependence of photocurrents on light irradiation. The resulting curves show an approximately exponential saturation behaviour. The correlation of this measurement to the saturated process of photosynthetic electron transfer itself may point to a direct link between the levels of photo-electrons available inside the cell, e.g. as the ratio of NADPH/NADP$^+$ molecules, and the peak number of photo-electrons measured in a BPV device. The magnitude and temporal variation of photocurrent generation is not yet well understood, and would depend on a complex interplay of biochemical processes, including competition with the terminal oxidases, which act as electron sinks[15]. Further research into the role of pigment concentration, saturation of light absorption and the regulation of terminal electron sinks within the cell will provide valuable insights into biophotovoltaics, as well as into the bioenergetics of photosynthesis in general. The quantitative data sets from the fast-response bio-anodes described here could provide an important contribution towards this goal.

The high electrochemical sensitivity of the porous anodes, particularly the microporous electrode enabling biofilm penetration, also enabled a sensitive real-time detection of the oxygen levels present inside photosynthetically active biofilms by CV (Fig. 7). Such analyses might provide quantitative insights into oxygenation studies e.g. in mixed biofilms or tissues[16].

Surprisingly, the performance of nanoporous and microporous electrodes was similar for both cyanobacteria. This similarity simplifies design rules of electrodes, requiring simply the sintering of a nanoparticle paste for the manufacture of nanoporous layers instead of more elaborate structuration. It also implies that

the dominating extracellular electron transfer mechanism may be self-mediated (no additional electron shuttling molecules were added). If the cyanobacteria were able to inject electrons efficiently into the electrode surface via DET, a larger performance increase would be expected for the microporous electrode, because it provides a many-fold increase in organism-electrode contact area compared to the two films. The potential DET-related performance increase was confirmed by using non-photosynthetic bacterium *Shewanella oneidensis*, which is well known to exhibit DET[13] and for which microporous bio-anodes showed a clear (ca. 5-fold) additional advantage over nanoporous films. DET tends to be fast, while avoiding the diffusion losses associated with soluble electron carriers[10], and can provide an important contribution to currents collected from electroactive biofilms. So-called 'nanowires' have been proposed to give rise to an efficient DET mechanism in some microbial biofilms. Nanowires are conductive extracellular pili-like structures that transport redox-electrons from within the cell through the insulating extracellular matrix and neighbouring biofilm cells to electrodes[13, 17], but their electrochemical properties are still disputed[18]. Conductive extracellular pili-like structures (PLS), ca. 10 μm in length, have recently been identified in the cyanobacteria utilised in this study, *Synechocystis* sp. PCC 6803 and *Nostoc punctiforme*, as well as in the cyanobacterium *Microcystis aeruginosa*, and it was speculated that they contribute to extracellular electron transfer in these organisms[13, 19, 20]. According to this hypothesis, the filaments should lead to a strong enhancement in photocurrents when brought in contact with the electrode surface across the entire photosynthetic biofilm, by using the microporous structure. The performance similarity of biofilms on microporous and nanoporous electrodes, however, suggests that the potential presence of conductive bacterial filaments did not contribute significantly to the photo–electron export in our devices, because the filaments did not connect to the organism's photosynthetic electron transfer chain or to the electrode, or the filaments were not produced. The measurements presented here suggest a slight advantage of the microporous electrode morphology over the nanoporous film. This advantage might be due to a larger, more accessible electrode surface, a smaller average distance self-mediated small molecules have to travel to reach the electrode, or the fact that biofilms reach further into the microporous electrodes, which, therefore, probe the photo response of bacteria that are exposed to high irradiation levels, and thus contain a larger number of available photo-electrons. On the other hand, the performance of microporous electrodes may be negatively affected by increased scattering of the inverse-opal structure, which leads to slightly increased reflection and decreased transmission, as well as an increase in electrode resistance with increasing thickness of the electrode.

It is clear that use of porous translucent electrodes offers a dramatic increase in the current density obtainable from photosynthetic microorganisms in biophotovoltaic devices. Their rapid photo-response times may additionally allow these devices to be exploited as a wider tool for the study of photosynthetic electron transfer.

## Methods
**Making of inverse-opal ITO electrodes**. The microporous electrodes were templated by polystyrene microspheres with an average diameter of 40 μm (Dynoseeds TS 40 from Microbeads). For each electrode, 450 μl of a 5%wt suspension in deionised water were dried on a conductive FTO-glass substrate (8 Ohm/sq, Sigma Aldrich). A homogeneous area (ca. 2 cm$^2$) of closed-packed spheres was achieved when drying the suspension at 30 °C on the substrate between two tightened aluminium sheets sandwiching a silicone O-ring, and with a hole in the upper sheet of the same size as the O-ring inner diameter. The frame can be used for convective assembly of colloids, but the large microspheres here settled on the substrate

quickly, decreasing the influence of meniscal forces on the assembly. The bead structure was annealed for 10 min at 130 °C on a hotplate. The nanoparticles were derived from a commercial ITO nanoparticle dispersion (<100 nm) 30%wt in isopropanol (IPA) (Sigma Aldrich) by drying and re-suspending in absolute ethanol (Sigma Aldrich) to result into a 10%wt suspension. Ethanol was better suited as a solvent because of its wetting properties with the polystyrene opal. IPA leads to an increased coverage of the template top, leaving fewer entry points for the microorganisms. The polystyrene opal was placed on a hotplate at 45 °C for quick drying and filled with 3 times with 25 μl of ITO suspension. The template was burned out on a hotplate inside a fume hood by heating the anodes to 500 °C, which simultaneously sinters the nanoparticles, finishing the electrode. For this purpose, the following heat ramp procedure was used: 3 h heat ramp from room temperature to 300 °C holding the temperature for 1 minute; 10 min ramp to 325 °C holding for 5 min; 10 min ramp to 375 °C for 5 min; 10 min to 450 °C for 15 min; 10 min to 500 °C for 15 min; off. This method can be found in detail on https://www.docubricks.com/viewer.jsp?id=22282785798926336.

For impedance spectroscopy testing, microporous electrode samples were prepared on ITO-coated glass substrates (to avoid an additional material interface) with the same protocol and annealed. They were then placed on upside down on a second ITO-glass substrate, which was freshly blade-coated with ITO nanoparticle paste (see below), and annealed again to sinter the paste and provide a good contact. After the annealing, the electrodes were filled with room temperature curing epoxy (EPO-TEK optical epoxy, 301-1LB kit) to provide handling stability. Four samples containing 130–150 μm thick microporous ITO layers made from the same nanoparticles had resistance values of 110–165 Ω.

**Making of flat ITO electrodes**. Plain ITO on PET electrodes with a surface resistivity of 100 Ω sq. were used as purchased from Sigma Aldrich. Nanoparticle film electrodes were made from the same commercial ITO nanoparticle dispersion and FTO-glass substrate as the microporous electrodes. 2.5 g equivalent of nanoparticles were mixed with 10.7 ml terpineol (a-Terpineol 96 + %, SAFC supply solutions). The IPA was evaporated off in a rotary evaporator at 55 °C and vacuum pumping. FTO glass was taped with Kapton tape (polyimide tape from RS components) as distance spacer, and the ITO-particle-terpineol paste was applied in between the tape strips. Excess paste was removed by manually pushing the side of a glass pipette rod over the spacers (blade-coating) to yield a plain film. The electrode was left to settle at room temperature for ca. 20 minutes. The Kapton tape was removed manually. The electrodes were annealed with the same heat ramp protocol described for microporous samples. This method can be found in detail on https://www.docubricks.com/viewer.jsp?id=22282785798926336.

For impedance spectroscopy testing, ITO nanoparticle films were prepared on commercial ITO-coated glass substrates (to avoid an additional material interface) with the same blade-coating protocol, and via spin-coating at different speeds. Commercial ITO-coated glass samples without any nanoparticle coating were annealed as the other samples, to provide an appropriate reference point despite the deterioration of ITO conductivity during the high temperature treatment. The mechanical junction of two individually annealed film samples provided a less reliable contact than the sintered microporous films, possibly causing a slight overestimation of their ohmic resistivity.

**Characterisation of porous electrodes**. The transmission spectra in Supplementary Fig. 3 of the film were recorded with a Varian Cary 300 UV-Vis spectrophotometer, and with an Ocean Optics FOIS-1 integrating sphere together with an Ocean Optics USB4000 spectrometer; both using uncoated FTO-glass as optical reference measurement (blank). The sheet resistivity of the nanoparticle film electrodes was determined on a non-conductive glass substrate with four point probe (S302 Lucas Labs and Keithley 2400) to be 100 ± 10 Ohm cm$^{-2}$. The thickness of the films was found to be 8.9 ± 0.6 μm by scanning over scratches with a Dektak 6 M stylus profiler. To probe the conductivity laterally, and through the depth of the electrode film and compare these values to the microporous inverse-opal structure, sandwich samples were built either by clipping two nanoparticle film samples on ITO-glass substrate on top of each other with two paper-clips, or by sintering a microporous electrode onto a film (details in respective electrode making method sections). The overlap area probing the material films was about 0.25 cm$^2$. The clean end pieces of the ITO-glass substrates were contacted with silver paste and crocodile clips, and electrode impedance spectroscopy measurements were performed across the enclosed films, using the following parameters on a Biologic SP-300 Potentiostat: scan at 0 V from 1 MHz to 1 Hz, with 40 points per decade, a sinus amplitude of 5 mV, waiting 0.1 periods before each frequency. No imaginary parts of the Nyquist plot were found that could indicate a non-ohmic resistance behaviour of the nanoporous or microporous films. The presented resistance data in this study are thus simply an average of the (noisy, but stable) real part of the measurements.

**Bacteria growth and quantification**. *Synechocystis* sp. PCC 6803 and *Nostoc punctiforme* cells were routinely cultured in BG11 medium supplemented with about 1 mM NaHCO$_3$ and maintained in sterile conditions at 30 °C under continuous moderate light of 40–50 μmol photons m$^{-2}$ s$^{-1}$ and shaking at 160 rpm.

Before applying cells to electrochemical devices, they were concentrated via centrifugation (ca. 2000 g for 10 min). A small volume of cells (120–160 µl) containing 134 nmol chlorophyll  was pipetted into the devices pre-filled with BG11 medium and after un-inoculated reference measurements. The heavy cell suspension quickly settled onto the anode to form a biofilm before the cell suspension can mix with the bulk electrolyte. The chlorophyll concentration was measured by extracting it from the cell suspension in 99.8% methanol (Sigma-Aldrich) and then calculating the chlorophyll-a concentration from two optical densities, as described previously[21].

**Biophotovoltaic devices**. The devices were printed on a Projet 3500HD Max 3D printer with Visijet M3X acrylic material and an accuracy of 0.025 mm. The support material (VisiJet S300) was washed out with hot sunflower oil and then IPA. Upright air-cathode windows (carbon paper with Pt-nanoparticles, Alfa Aesar 45372 Hydrogen Electrode/Reformate) were sealed into the 3D-printed device with dental silicone (Zhermack, Elite HD + Super Light Body). The anodes were tightened in the device manually with screws and sealed with a nitrile O-ring. A good contact between the anode edges and the connector-clamp was ensured by conductive silver paste.

**BPV operation and measurements**. The BPV devices were loaded with the different anode types and filled with a 10 ml volume of BG11 medium (containing small amounts of phosphate) and phosphate buffer (DPBS 10 × , D1408 Sigma-Aldrich, pH 7.2) in a mixing ratio of 9:1.07 to obtain an overall phosphate buffer concentration of 10 mM. Experiments referred to as 'no added phosphate buffer' used devices filled only with BG11 medium. The electrochemical measurements were performed with a potentiostat (MultiEmStat by PalmSens) with 12 independent channels. The potentiostat applies a voltage and measures the resulting current with a high resolution of 1 nA. Reference measurements were conducted with the connected devices after at least 6 hour waiting time (when chronoamperometry measurements seemed to have reached equilibrium) and before adding cyanobacteria. Then, 134 nmol chlorophyll-a equivalent of cyanobacteria were added to each device (see Bacteria growth and quantification). Measurements were conducted again after a settling time of at least 8 h and usually for a duration of one to three days. The illumination periods with high light intensities were limited to five or seven minutes, while more moderate photon flux densities of ca. 500 µmol m$^{-2}$ s$^{-1}$ were used for longer exposures. In the case of the custom LED white light source (Supplementary Fig. 4), the moderate illumination compares to the world average sun intensity reaching ground level within the visible light spectrum. The temperature of the devices was held stable at 25 °C by water cooling or heating of the hollow devices with an external water bath.

The number of measurement channels per anode type used to obtain the presented averages and standard deviations vary slightly because the devices that shorted due to water condensation or leakage were not considered. During BPV device measurements with *Synechocystis*, all four devices each with microporous and nanoporous electrodes and three devices with non-porous electrodes delivered uninterrupted data. In half-cell mode, averages were formed of three microporous electrode bearing devices, four with NP-films and three with non-porous electrodes. For *Nostoc* measurements, the numbers were two, four and two, respectively. No increase in *Nostoc* electro-activity was observed after several days in the device, in contrast to the literature[20].

**Shewanella fuel-cell measurements**. Measurements on *Shewanella oneidensis* bacteria were conducted in half-cell mode at an anode potential of 0.2 V vs. an Ag/AgCl reference electrode in fresh LB medium without additional phosphate buffer. The same amount of concentrated cell suspension was added to each device (1400 µl each, cells were separated by repeated pipetting with a 1 ml tip), and transmission spectra were recorded at different cell densities as a point of reference for cell numbers, see Supplementary Fig. 11. Data were generated from four devices with microporous and nanoporous electrodes each, and two non-porous ITO electrodes. The devices were covered with parafilm before measuring, which reduced the oxygen supply. The parafilm did not create fully anaerobic conditions, since during CV measurements the presence of residual oxygen influx could still be detected.

**Microscopy**. Optical microscopy was conducted with an Olympus BX60 microscope and Olympus UMPlanFl 20× and 100× objectives. Images at manually adjusted different focal depths were taken with an AxioCam MRc 5 (Zeiss) camera and Z-stacked with CombineZP software. Slight adjustments were made to the colour balance and contrast of the image. For electron microscopy images, a Leo Gemini 1530 VP SEM was used with a Schottky-emitter consisting of a zirconium oxide coated tungsten cathode and an in-lens secondary electron detector.

**Data availability**. Data associated with research published in this paper can be accessed at https://osf.io/e7g9f/.

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

## Acknowledgements

We thank Antonio Abate for advice on conductivity testing across the films as well as interpreting the impedance spectroscopy results and Erika Eiser for discussion and the provision of colloids. T.W. is grateful for funding from the Winton Fund for physics of sustainability, Mott Fund, Cambridge Trust (CHESS) and the EPSRC CDT in

Nanoscience and Nanotechnology. D.H. thanks Evangelisches Studienwerk—Villigst e.V. for funding during this work. P.B. and C.J.H. thank the Leverhulme Trust for financial support. U.S. acknowledges funding from the Adolphe Merkle Foundation.

## Author contributions

T.W., U.S., P.B. and C.H. conceived the project and designed the experiments. T.W. collected and analysed all included data and performed imaging. T.W. & D.H. designed and built electrochemical devices, light source and method for microporous electrodes with advice from P.B. & U.S. T.W. wrote the manuscript, further improved by all authors.

## Additional information

**Competing interests:** The authors declare no competing interests.

