## [Peer Review File · Nature Communications]

Reviewers' comments:

Reviewer #1 (Remarks to the Author):

The authors explore the effects of electrode porosity on the biophotovoltaic behavior of cyanobacteria. Non-porous, nanoporous, and microporous ITO-based anodes are used in the construction of biophotovoltaic devices, and the resulting photocurrents are compared. The porous electrodes show an enhanced photocurrent density, and peak current is shown to increase with increasing light intensity until saturation. Photoelectrochemical characterizations are accompanied by optical and electron microscopy images used to characterize electrode morphology. In their characterizations, the authors show photocurrent density (normalized by area), CV measurements, illumination light source characterization, and measurements for multiple devices. These are rigorous techniques central to biophotovoltaic measurements that still need to be standardized and adapted by the community.

Since this study offers significant insight in the area of electrode effects and transport mechanism (DET vs. MET), an area that remains elusive in the field, publication is recommended. However, two critical points should be addressed prior to publication. First, though the photocurrent is normalized by surface area, it is unclear as to how the authors determined the surface area. In the photovoltaic community, electroactive surface area (as determined by electrochemical characterization of the electrode in the presence of a mediator with known redox characteristics, such as ferricyanide) is often used to determine current density. The advantage of this approach is that it accounts for the difference in surface area due to morphology (nano- vs. micro-porous), which would allow the authors to accurately evaluate the effects of DET.

The second concern lies with the presentation of the control measurements in the study. Initially, it appears that the manuscript fails to address multiple key concerns. For example, the supplemental data show that the porous electrodes show higher photocurrent than non-porous electrodes even in the absence of cells (Fig. S7) and the microporous electrodes are less transparent than the other electrode (Fig. S3). These findings would suggest that the enhanced photocurrent observed for the porous electrodes could be due to an inherent electrode mechanism (as opposed to bacteria-electrode interaction) or that the collected current from the macroporous electrode is underestimated (since the illumination intensity is less). Underestimation of the macroporous electrode current would have a significant impact on the findings of this manuscript, since its value compared to that of the nanoporous electrode determines whether DET or MET is dominating. The control measurements with the *Shewanella* and *Nostdoc* cells should be able to largely address these concerns. *Shewanella* offers an important control vs. light effects whereas *Nostdoc* offers an important control vs. morphological effects by offering a comparison of unicellular vs. filamentous bacteria effects. However, these observations are not explicitly addressed in the manuscript, and the significance of the *Nostdoc* and *Shewanella* measurements are underplayed.

A few minor recommendations are also listed below:

1. A few key relevant studies on the topic of extracellular electron transfer in cyanobacteria and *Shewanella* are not referenced in the study. Schuergers et al. (Energy & Environmental Science 10, 1102-1115, 2017) offer a direct quantitative basis for comparison of maximum efficiency based on the Gibbs free energy change (or similarly, maximum voltage) for *Shewanella* and cyanobacteria devices. This reference is quite relevant to the discussion on page 2 of the manuscript. In addition, on page 8, the authors discuss microbial DET through conductive nanowires. However, these observations are currently debated in the field, and this debate should be explicitly acknowledged in the manuscript (see Yates, et al. Nature Nanotechnology 11, 910-913, 2016).

2. The band diagram in Fig. 1 is not intuitive to follow. It would help to explicitly identify the 1.23 V (vs. SHE) level for water-splitting and maintain consistent labeling in the left and right vertical axes ("Energy" on the left and "E" on the right). The authors should include a discussion on alternative redox reactions that could occur on the cathode (for example, the direct recombination of H⁺ to produce H₂).
3. Microporous measurements are not shown in Fig. S2, although they are discussed in the caption.
4. Fig. S5 y-axis is labeled as current difference and not current. Light intensities for each cycle area not specified, and it is unclear as to the same light intensity penetrates through the ITO since micro- and nano-porous have different absorbance.
5. Fig. 4 caption should mention that the device is run under short-circuit conditions, and the potentiostat connection should be included in the schematic. Nostdoc data are also not shown.
6. The authors mention device efficiency for Fig. 6, but its calculation is not discussed.
7. The authors state that the non-porous electrodes performed better (page 6), though this is not directly clear in Fig. S7.
8. The transient spikes observed in the chronoamperometry measurements when the light is turned off suggests charge accumulation. This observation is consistent with the surface charging effects seen in the CV and can be used to strengthen the authors' argument in this regard.
9. The author should discuss the differences in the shape of the curves observed for non-porous vs. nanoporous and *Synechocystis* vs. *Nostdoc* curves in Fig. 5.
10. Overall, the manuscript is well written and easy to follow. There are a few typographical errors, particularly in the methods and supplement.

Reviewer #2 (Remarks to the Author):

Wenzel et al. describe the use of porous ITO electrode to enhance the currents generated by biofilms of photosynthetic cyanobacteria. The use of porous ITO over flat electrodes increases the currents by two orders of magnitude. The authors give detailed information about the electrode fabrication process, demonstrate 3D printed devices, and give report detailed measurements with good controls. I find their results very interesting and suitable for publication. I have only minor suggestions.

On page 6, it is not clear to me what is meant with "...also stronger surface charging, leading to high faradaic currents...". Should a larger surface not simply lead to larger capacitive currents?

The photoresponse of the biofilm is quite complex, in, for instance, fig S5 the current decreases before and after the light source is switched on. What is the reason for these significant changes in current?

In the discussion section, the authors suggest that conductive nanowires could be important for their high observed efficiencies. Can they provide additional information on why this could be the case, do they plan to prove this hypothesis?

Dear Editor,

Thank you very much for the review and your feedback.

We are gratified by the positive reviewer response. The issues they have raised are straight forward to address and we have incorporated them into the revised version of the manuscript. For your convenience, we are submitting a version of the manuscript in which all changes are made.

We hope that this will clear the way for the publication of our manuscript.

With best wishes,

Ulli Steiner

Reviewers' comments:

Reviewer #1 (Remarks to the Author):

The authors explore the effects of electrode porosity on the biophotovoltaic behavior of cyanobacteria. Non-porous, nanoporous, and microporous ITO-based anodes are used in the construction of biophotovoltaic devices, and the resulting photocurrents are compared. The porous electrodes show an enhanced photocurrent density, and peak current is shown to increase with increasing light intensity until saturation. Photoelectrochemical characterizations are accompanied by optical and electron microscopy images used to characterize electrode morphology. In their characterizations, the authors show photocurrent density (normalized by area), CV measurements, illumination light source characterization, and measurements for multiple devices. These are rigorous techniques central to biophotovoltaic measurements that still need to be standardized and adapted by the community.

Since this study offers significant insight in the area of electrode effects and transport mechanism (DET vs. MET), an area that remains elusive in the field, publication is recommended. However, two critical points should be addressed prior to publication.

We thank the reviewer for his useful comments and his positive recommendation

(1) First, though the photocurrent is normalized by surface area, it is unclear as to how the authors determined the surface area. In the photovoltaic community, electroactive surface area (as determined by electrochemical characterization of the electrode in the presence of a mediator with known redox characteristics, such as ferricyanide) is often used to determine current density. The advantage of this approach is that it accounts for the difference in surface area due to morphology (nano- vs. micro-porous), which would allow the authors to accurately evaluate the effects of DET.

For normalisation, the macroscopic illuminated photo-electrode surface area was used, as is typical for light harvesting devices. In this macroscopic measure, the micro- and nano-structure of the electrode is not taken into account, as it does not extend the illumination area. Our electrodes have an area of 1cm^2 , as defined by the O-ring that was used and the opening in the printed devices. We have further clarified this in the revised manuscript, under the "Biophotovoltaic devices" section.

The amount of surface charging in porous anodes (shown in Fig. S8) gives an indication of the electrode surface area similar to the ferricyanide method suggested by the reviewer or even more

precise gas-sorption measurements. Our data exhibits comparable behaviour for both anode types (nanoporous and microporous), while the non-porous anode shows orders of magnitude lower charging. This is now mentioned in the discussion section. We have furthermore included an additional CV measurement in the supplementary information (Fig. S9) that confirms this conclusion.

Also, please note that we do not claim to have accurately quantified the direct electron transfer (DET) effect. The large (ca. five fold) difference of current observed from *Shewanella* bacteria on the microporous versus the nanoporous electrode are certainly associated with DET, but the minor photo-current differences between these two electrode types might be caused not only by small differences in surface area, but also by increasing resistance with height, light scattering, and distance of bacteria film to substrate electrode. Given our biological references, we can only say that in our wildtype cyanobacteria, DET does not seem to provide a photo-current contribution large enough to be detected in our experiment (and thus to play an important role in energy harvesting devices).

(2) The second concern lies with the presentation of the control measurements in the study. Initially, it appears that the manuscript fails to address multiple key concerns. For example, the supplemental data show that the porous electrodes show higher photocurrent than non-porous electrodes even in the absence of cells (Fig. S7) and the microporous electrodes are less transparent than the other electrode (Fig. S3). These findings would suggest that the enhanced photocurrent observed for the porous electrodes could be due to an inherent electrode mechanism (as opposed to bacteria-electrode interaction) or that the collected current from the macroporous electrode is underestimated (since the illumination intensity is less). Underestimation of the macroporous electrode current would have a significant impact on the findings of this manuscript, since its value compared to that of the nanoporous electrode determines whether DET or MET is dominating. The control measurements with the *Shewanella* and *Nostoc* cells should be able to largely address these concerns. *Shewanella* offers an important control vs. light effects whereas *Nostoc* offers an important control vs. morphological effects by offering a comparison of unicellular vs. filamentous bacteria effects. However, these observations are not explicitly addressed in the manuscript, and the significance of the *Nostoc* and *Shewanella* measurements are underplayed.

We thank the referee for this helpful suggestion to increase the clarity of our manuscript. Background currents were subtracted from each experiment. The signals were comparable between the two electrode types (Fig. 4) for both *Synechocystis* and *Nostoc* cyanobacteria, indicating that the surface area is large enough in both electrodes to collect electrons sensitively. The similarity of saturation curves for these electrode types (Fig. 6) also suggests that shading due to light scattering are small (as expected from an optical analysis). The *Shewanella* measurements confirm that the direct contact of bacteria with the electrode surface is enhanced in microporous electrodes, because the known direct electron transfer in those bacteria leads to much higher currents compared to the nanoporous electrode.

We have now clarified these points in the “Light response characteristics” results section.

A few minor recommendations are also listed below:

1. A few key relevant studies on the topic of extracellular electron transfer in cyanobacteria and *Shewanella* are not referenced in the study. Schuergers et al. (*Energy & Environmental Science* 10, 1102-1115, 2017) offer a direct quantitative basis for comparison of maximum efficiency based of the Gibbs free energy change (or similarly, maximum voltage) for *Shewanella* and cyanobacteria

devices. This reference is quite relevant to the discussion on page 2 of the manuscript. In addition, on page 8, the authors discuss microbial DET through conductive nanowires. However, these observations are currently debated in the field, and this debate should be explicitly acknowledged in the manuscript (see Yates, et al. Nature Nanotechnology 11, 910-913, 2016).

Thank you for suggesting these suitable references. We have incorporated them in the manuscript.

2. The band diagram in Fig. 1 is not intuitive to follow. It would help to explicitly identify the 1.23 V (vs. SHE) level for water-splitting and maintain consistent labeling in the left and right vertical axes ("Energy" on the left and "E" on the right). The authors should include a discussion on alternative redox reactions that could occur on the cathode (for example, the direct recombination of H⁺ to produce H₂).

We have unified the labelling of the diagram and added the hydrogen generation level as an alternative cathode reaction. The water splitting level at neutral pH was already included in the diagram (in hydrolysis, it only shifts to 1.23V at 0 pH, where the hydrogen generation level lowers to 0V).

3. Microporous measurements are not shown in Fig. S2, although they are discussed in the caption.

The mention of microporous electrodes was removed and the details were transferred to the Methods section. The mentioning in the caption was redundant as the statement already occurs in the text where the figure is introduced.

4. Fig. S5 y-axis is labeled as current difference and not current. Light intensities for each cycle area not specified, and it is unclear as to the same light intensity penetrates through the ITO since micro- and nano-porous have different absorbance.

The figure was improved as suggested. The photon flux values (incoming, just above the bioanodes) are now indicated. The initial dark-current of the data is no longer subtracted and so the axis label could be changed from "Current difference" to "Current".

This figure (S5) shows an example of a chrono-amperometry trace and how peak values were determined. The dynamic was not affected by the micro-pores.

5. Fig. 4 caption should mention that the device is run under short-circuit conditions, and the potentiostat connection should be included in the schematic. Nostdoc data are also not shown.

The connections, and explanations have been added for clarity as suggested. The Nostoc data is already shown in figure 6 in detail.

6. The authors mention device efficiency for Fig. 6, but its calculation is not discussed.

An explanation has been added on page 6.

7. The authors state that the non-porous electrodes performed better (page 6), though this is not directly clear in Fig. S7.

A label has been added to the photocurrent bars in figure S7 to indicate that the difference factor there is much smaller than 300, but still larger than 10.

8. The transient spikes observed in the chronoamperometry measurements when the light is turned off suggests charge accumulation. This observation is consistent with the surface charging effects seen in the CV and can be used to strengthen the authors' argument in this regard.

Thank you. We included this indication in the argument on page 6.

9. The author should discuss the differences in the shape of the curves observed for non-porous vs. nanoporous and *Synechocystis* vs. *Nostdoc* curves in Fig. 5.

The current dip in *Synechocystis* data on non-porous electrodes appears to be a singular event for the specific presented measurement set that has not occurred during our other measurements (which were however recorded in a slightly different fashion such as shorter or longer illumination times). We have therefore not given further importance to the dynamic, but have now indicated this accordingly on page 6.

10. Overall, the manuscript is well written and easy to follow. There are a few typographical errors, particularly in the methods and supplement.

Thank you. We have corrected those errors we could identify in this version (not highlighted in tracked-changes-document). Some sentences have been exchanged for added clarity.

Reviewer #2 (Remarks to the Author):

Wenzel et al. describe the use of porous ITO electrode to enhance the currents generated by biofilms of photosynthetic cyanobacteria. The use of porous ITO over flat electrodes increases the currents by two orders of magnitude. The authors give detailed information about the electrode fabrication process, demonstrate 3D printed devices, and give report detailed measurements with good controls. I find their results very interesting and suitable for publication. I have only minor suggestions.

1. On page 6, it is not clear to me what is meant with "...also stronger surface charging, leading to high faradaic currents...". Should a larger surface not simply lead to larger capacitive currents?

Thank you for identifying this error. It was supposed to read non-Faradaic currents and has now been clarified in the manuscript.

2. The photoresponse of the biofilm is quite complex, in, for instance, fig S5 the current decreases before and after the light source is switched on. What is the reason for these significant changes in current?

The main reason for these transient peaks immediately after switching the light on and off is charge accumulation. The accumulated electrons during illumination lead to this polarisation behaviour when the light is switched off. This has been added to manuscript page 6 in the context of the (non-Faradaic) charging currents as also suggested by reviewer1. The charging that appears when the light is switched on could be a more interesting behaviour e.g. related to ion release. We intend to investigate this very brief (light-on) charging effect in a separate study.

3. In the discussion section, the authors suggest that conductive nanowires could be important for their high observed efficiencies. Can they provide additional information on why this could be the case, do they plan to prove this hypothesis?

Conductive “nanowires” do not only have the efficiency advantages of direct electron transfer (now specified as “while avoiding the diffusion losses associated with soluble electron carriers”), they can also transfer electrons over larger distances. E.g. in our microporous electrode not every cell directly touches the electrode. Nanowires of several micrometre length would enable additional DET electron transfer (now specified as “(... matrix) and neighbouring biofilm cells”). This effect has been demonstrated in *Geobacter* biofilms in the two respective references. If we find an example where conductive filaments participate in the electron transfer of photosynthetic microorganisms, we will certainly attempt to characterise this effect in detail.

REVIEWERS' COMMENTS:

Reviewer #1 (Remarks to the Author):

The authors have addressed the points raised in my previous review, and publication of the manuscript as is recommended. One minor point in the new dataset from Figure S9 that is a bit unclear is that the measurements indicate that both the nanoporous and macroporous electrodes undergo a similar extent of surface charging, when, in theory, the nanoporous electrode should have a larger surface area.

Below I have included a few suggestions that could further clarify the manuscript.

1. Figure 1 – specify in the caption that values are for physiological pH to avoid confusion.
2. Figure 6 – caption states “Nostdoc biofilm peak photocurrents” whereas text reads “For...Nostdoc measurements, irradiation levels were not long enough to reach full peak values”. It is recommended to adjust the caption to “Nostdoc biofilm maximum photocurrents” to be consistent with the text.
3. Figure S7 – indicate in the caption that the shaded regions are standard deviations. There also seems to be some confusion from my previous comment regarding this figure. In the sentence, “...non-porous ITO electrodes performed better and responded faster in the absence of phosphate buffer.”, it is recommended to add “...compared to relative performance in the presence of buffer.” to clarify that the comparison is made with and without buffer.
4. Figure S9 – change “extend” to “extent”

Reviewer #2 (Remarks to the Author):

The authors have addressed all my concerns and therefore I support publication without further changes

RESPONSE TO REVIEWERS:

Reviewer #1 (Remarks to the Author):

The authors have addressed the points raised in my previous review, and publication of the manuscript as is recommended. One minor point in the new dataset from Figure S9 that is a bit unclear is that the measurements indicate that both the nanoporous and macroporous electrodes undergo a similar extent of surface charging, when, in theory, the nanoporous electrode should have a larger surface area.

We have added a reminder in the capture of Figure S9 that our macroporous electrodes are at the same time also nanoporous, since they consist of the same nanoparticles. For this reason, a similar surface charging (as observed) is to be expected.

Below I have included a few suggestions that could further clarify the manuscript.

1. Figure 1 – specify in the caption that values are for physiological pH to avoid confusion.

Done.

2. Figure 6 – caption states “Nostdoc biofilm peak photocurrents” whereas text reads “For...Nostdoc measurements, irradiation levels were not long enough to reach full peak values”. It is recommended to adjust the caption to “Nostdoc biofilm maximum photocurrents” to be consistent with the text.

OK, thank you for the helpful suggestion.

3. Figure S7 – indicate in the caption that the shaded regions are standard deviations. There also seems to be some confusion from my previous comment regarding this figure. In the sentence, “...non-porous ITO electrodes performed better and responded faster in the absence of phosphate buffer.”, it is recommended to add “...compared to relative performance in the presence of buffer.” to clarify that the comparison is made with and without buffer.

Both done, thank you.

4. Figure S9 – change “extend” to “extent”

Done.

Reviewer #2 (Remarks to the Author):

The authors have addressed all my concerns and therefore I support publication without further changes

Thank you!